# DNA-Based Method for Traceability and Authentication of *Apis cerana* and *A. dorsata* Honey (Hymenoptera: Apidae), Using the *NADH dehydrogenase 2* Gene

**DOI:** 10.3390/foods11070928

**Published:** 2022-03-23

**Authors:** Saeed Mohamadzade Namin, Fatema Yeasmin, Hyong Woo Choi, Chuleui Jung

**Affiliations:** 1Agricultural Science and Technology Institute, Andong National University, Andong 36729, Korea; saeedmn2005@gmail.com; 2Department of Plant Protection, Faculty of Agriculture, Varamin-Pishva Branch, Islamic Azad University, Varamin 3381774895, Iran; 3Department of Plant Medicals, Andong National University, Andong 36729, Korea; fatema.setudu@gmail.com (F.Y.); hwchoi@anu.ac.kr (H.W.C.)

**Keywords:** honey, entomological origin, mitochondrial DNA, *NADH dehydrogenase 2*, PCR

## Abstract

Honey is a widely used natural product and the price of honey from *Apis cerana* (ACH) and *A. dorsata* (ADH) is several times more expensive than the one from *A. mellifera* (AMH), thus there are increasing fraud issues reported in the market by mislabeling or mixing honeys with different entomological origins. In this study, three species-specific primers, targeting the *NADH dehydrogenase 2* (*ND2*) region of honeybee mitochondrial DNA, were designed and tested to distinguish the entomological origin of ACH, ADH, and AMH. Molecular analysis showed that each primer set can specifically detect the *ND2* region from the targeted honeybee DNA, but not from the others. The amplicon size for *A. cerana*, *A. dorsata* and *A. mellifera* were 224, 302, and 377 bp, respectively. Importantly, each primer set also specifically produced amplicons with expected size from the DNA prepared from honey samples with different entomological origins. The PCR adulteration test allowed detection of 1% of AMH in the mixture with either ACH or ADH. Furthermore, real-time PCR and melting curve analysis indicated the possible discrimination of origin of honey samples. Therefore, we provide the newly developed PCR-based method that can be used to determine the entomological origin of the three kinds of honey.

## 1. Introduction

Honey is a sweet natural product produced by honey bees using the nectar, secretions of living parts, or honeydew of plants [1,2]. Due to the broader geographical distribution, *Apis mellifera* honey (AMH), *A. cerana* honey (ACH), and *A. dorsata* honey (ADH) are the three dominant types of honey in the Asian market. Giant honeybee (*Apis dorsata* F.) is distributed throughout South and Southeast Asia and China [3]. The colonies of *A. dorsata* are generally found in rainforests or on the cliffs, but they also can be occasionally found in building ledges of urban areas. Even though *A. dorsata* is not domesticated and cannot be maintained for honey harvest or pollination purposes, it plays an important role in the pollination of tropical rainforest plants and local crops [4,5,6]. Due to the fact that *A. dorsata* is considered the most defensive honeybee compared to other *Apis* spp., ADH is harvested by highly motivated experts (so-called honey hunters) [7]. Furthermore, ADH contains the highest concentration of phenolic compounds and flavonoids compared to other honeys, thus exhibiting high DPPH (2,2-diphenyl-1-picrylhydrazy) free radical-scavenging activity, FRAP (Ferric reducing-antioxidant power assay) values and the lowest AEAC (Ascorbic acid Equivalent Antioxidant Capacity) values, as well. This indicates that ADH has strong antioxidant properties and medical values [8,9]. Asian honeybee (*A. cerana* F.) is widespread in South, South East, and Eastern Asia from Afghanistan to Far East Russia and Japan [10]. It is one of the domesticated honeybees; however, due to its lower productivity compared to *A*. *mellifera*, most beekeepers prefer *A. mellifera* over *A. cerana* [11]. In addition, high interspecific competition between *A. mellifera* and *A. cerana* on the same niche resulted in the decline of *A. cerana* colonies in many countries such as China and Korea in the last decades [11,12,13]. This led to lower production of ADH and ACH compared to AMH.

Although honey is one of the most widely consumed natural products, it is one of the most counterfeited food products in the market [14]. Due to the fast growth of the human population and the rising demands toward the consumption of organic and local products, the entomological origin of honey has been taken into consideration. Therefore, the market price of ADH and ACH is several times higher than AMH. This situation makes ADH and ACH vulnerable to adulteration problems, either by mislabeling (claiming the false geographical, botanical, or entomological origin of honey) or by mixing (overfeeding the bees with sugars, adding sweeteners or syrups, and dilution with cheaper honey) in order to increase the economic profit [2,3,15,16,17,18,19].Thus, it is important to develop rapid, reliable, and cost-effective identification methods for the entomological origin of honey to solve the adulteration problem in the market.

Molecular detection of the entomological origin of honey by using the set of specific primers is regarded as a rapid, accurate, and suitable tool for the identification of the origin of animal products and processed foods [20,21,22,23]. Considering the method of processing honey by honeybees, the bee cells can remain inside of the honey. Given the opportunity to extract bee DNA from honey, it is possible to use it for the identification of the entomological origin of honey. Compared to the other identification methods for entomological origin of honey, such as SDS-PAGE or chemical-based methods [12,24,25], the DNA-based method is more precise, quick, and suitable for analysis of a large sample size [26]. Recently, several studies were conducted using DNA-based methods to identify the entomological origin of honey. Zhang et al. [26] developed a gDNA-based method for the identification of two different major honeys, AMH and ACH, in the market. Two sets of primers were designed to amplify *Major royal jelly protein 2* (*MRJP2*) gene, resulting in the different sizes of PCR product in the gel electrophoresis, making it useful to discriminate ACH from AMH. In addition, it is also possible to identify the honey samples through Real-Time PCR based on their melting temperature analysis [26].

Mitochondrial DNA (mtDNA) is present in most cells with high copy numbers. It is characterized by a high genetic variation between related species but a low intraspecific variation [27,28,29]. Therefore, it is suitable to use mtDNA for taxonomic and phylogenetic analysis. Targeting the *cytochrome oxidase I* (*COI*) gene of mtDNA, Kim et al. [30] designed species-specific primers to differentiate ACH and AMH. PCR with designed primer sets produced amplicons with a length of 133bp and 178 bp for *A. mellifera* and *A. cerana*, respectively. Although the size of the amplicons is distinguishable and even applicable for relatively old honey samples, Zhang et al. [26] reported the designed primers for *A. mellifera* was not species-specific as they made the same length of band from ACH-originated DNA extracts in China. Soares et al. [31] developed species-specific primers to amplify the intergenic region of tRNAleu-cox2, enabling the detection of *A. cerana* DNA using PCR. In addition, they discriminated ACH and AMH using high-resolution melting curve analysis targeting the 16S rRNA gene, making it possible to detect the entomological origin of ACH. However, the lack of species-specific primer designed for mtDNA of *A. mellifera* makes it difficult to use it for adulteration studies. The only species-specific primer set that is available to detect *A. mellifera* is provided by Zhang et al. [26]. However, the size of the PCR product (~560 bp) is largely applied to relatively old honey samples due to DNA degradation, and it is important to design species-specific primers targeting smaller regions. On the other hand, there is no species-specific primer available for reliable and cost-effective identification of the entomological origin of ADH.

In this study, we aimed to develop a rapid and accurate PCR-based method to recognize the entomological origin of ADH, ACH, and AMH. This method can also be applied to discriminate between pure and adulterated honey. We also aimed to provide species-specific primers targeting smaller parts of mtDNA to avoid the negative effect of possible DNA degradation, which may happen during the storage of honey. In this study, three species-specific primers for ADH, ACH, and AMH were designed to amplify the short part of the *NADH dehydrogenase 2* (*ND2*) region of the mtDNA. Our experiment suggests that species-specific primer sets targeting *ND2* not only successfully distinguished ADH, ACH,s and AMH, but also detected mixed 1% AMH from ADH or ACH. Additionally, several honey samples from different countries were used to evaluate the accuracy of the developed method.

## 2. Material and Methods

### 2.1. Schematic Overview of the Experimental Design

In this study, 3 species-specific primer sets were designed to test the traceability of the entomological origin of honey. The specificity and sensitivity of the primers were tested first with the DNA extracts from honeybees with different geographical origins. Then, the DNA extracts from artificially mixed honey samples were used to evaluate the applicability of using designed primer sets in honey authentication. Subsequently, the developed method was used to check the entomological origin of honey provided by honey hunters and beekeepers (Figure 1).

### 2.2. Designing Species-Specific Primers

*NADH dehydrogenase 2* (*ND2*) region of mitochondrial DNA was used as a target area. The complete mitochondrial genome sequence of *A. cerana*, *A. dorsata* and, *A. mellifera* were obtained from NCBI (Appendix A) and used for designing species-specific primers using OLIGO 7 primer analysis software (Table 1). The Primer-BLAST tool was initially used to determine primer specificity (http://www.ncbi.nlm.nih.gov/tools/primer-blast accessed on 25 March 2020). Designed primers were synthesized by Macrogen (Daejan, Korea).

### 2.3. Evaluate the Specificity and Sensitivity of Designed Primer Sets Using Bee DNA

-DNA extraction from honeybees

DNA extracts from adult or larvae samples of honeybees of *A. cerana* (5 adults from Nepal, 5 adults from Korea), *A. dorsata* (5 larvae from Thailand, 5 adults from Nepal) and *A. mellifera* (5 adults from Nepal, 5 adults from Korea) were used to test the specificity and sensitivity of the primer sets. Bee DNA was extracted from the left hind leg of adult honeybee or head and thorax part of larvae using DNeasy blood and tissue kit (Qiagen, Hilden, Germany) following manufacturer’s instruction.

-Specificity and sensitivitytest of designed primers

The DNA extracts from honeybees with different geographical origins were used to examine the specificity of the designed species-specific primers. The PCR procedure was carried out in a 20 µL reaction volume mixture containing 100 ng of template DNA and 1μL of each primer (10 pmole/μL) using AccuPower PCR PreMix (Bioneer, Daejan, Korea). The thermal cycling procedure contained an initial pre-denaturation at 95 °C for 5 min, and 35 cycles of 95 °C for 30 s, 52 °C for 30 s, 72 °C for 40 s, and a final extension of 72 °C for 5 min in a BIOER thermal cycler. 7 µL of PCR products were analyzed using 2.5% agarose gel in TAE buffer and the bands were visualized by EcoDye (BIOFACT) and gel document system (GSD-200D).

To evaluate the detection limit of the species-specific primers, DNA extracts from different bee samples were serially diluted by 10-fold (100 to 0.01 ng/μL) and used for PCR analysis.

-Melting curve analysis by real-time PCR

DNA extract from honeybees were used to evaluate the possibility of using real-time PCR-based detection of adulteration of ACH and ADH. The real-time PCR was carried out using 10 μL of 2X Real-Time PCR Master Mix (BioFACT) including SYBR Green I, 100 ng of DNA template and 1 µL of each primer (10 pmole/µL) in 20 µL of total reaction mixture. PCR cycling was as follows: 95 °C for 15 min, following 35 cycles of 95 °C for 30 s, 52 °C for 30 s and 72 °C for 40 s. For analyzing melt curve, real-time PCR products were denatured at 95 °C for 15 s, annealed at 52 °C for 1 min then followed by melting curve ranging from 52 to 95 °C with temperature increments of 0.3 °C every 20 s. The data of real-time PCR and melt curve analysis were processed using FQD-96a V1.0.13 software (BIOER, Hangzhou, China).

### 2.4. PCR-Based Sensitivity Test of Honey Samples

-Preparation of honey samples

Three pure honey samples (ACH and AMH from Korea and ADH from Thailand) were used to test the sensitivity of species-specific primers to detect honey adulteration. AMH was mixed with ACH or ADH in different proportions (100:0, 99:1, 95:5, 50:50, 20:180, 0:100), and then used for DNA extraction.

-DNA extraction from honey samples

To extract DNA from honey samples, 40 mL distilled water was added to 15 g of honey, incubated at 45 °C for 30 min, vortexed and centrifuged at 15,000 rpm for 30 min. The supernatant was discarded, and the pellet was dissolved in 1 mL of distilled water and centrifuged at 15,000 rpm for 15 min. The supernatant was discarded, and the pellet was used for DNA extraction using DNeasy mericon Food Kit (Qiagen, Hilden, Germany) following manufacturer’s protocol. The concentration and purity of the DNA extracts were evaluated using Nano Drop spectrophotometer (Life Real). Extracted DNA was used for the subsequent PCR analysis.

-Polymerase chain reaction (PCR)

For sensitivity (or adulteration) test, 2 rounds of PCR were performed with same primer sets. In the first round of PCR, 100 ng of DNA was used as a template DNA following the same protocol described above. Then, 5 μL of PCR product was used as a template DNA for the second round of PCR with the same protocol described above. A total of 7 µL of final PCR products were analyzed in 2.5% agarose gel. In addition, PCR using DNA extracts from honey samples (10 with ACH and 5 with ADH labels) was conducted for honey adulteration test using the species-specific primers. Two rounds of PCR were performed as described above. To confirm the amplified DNA sequence, PCR products were analyzed in 2.5% agarose gel, purified and sequenced by Macrogen (Daejan, Korea) using an ABI 3130xl capillary automated.

### 2.5. Adulteration Analysis of the Honey Samples

-Honey samples

The purity of 20 honey samples (10 ACH, 5 ADH and 5 AMH) from different localities (Nepal, Thailand and Korea) were evaluated. ACH samples were provided by beekeepers from Nepal (*n* = 2), Thailand (*n* = 5) and Korea (*n* = 3). ADH samples were provided by honey hunters from Thailand (*n* = 5). AMH samples were harvested directly by beekeepers from Korea (*n* = 5) (Appendix A). All samples were collected in 2020. Honey samples were stored at −20 °C and 4 °C prior to DNA extraction, respectively.

-DNA extraction and PCR-based authentication of honey samples

DNA was extracted from all honey samples using the method that was described before. PCR using DNA extracts from honey samples (10 with ACH and 5 with ADH labels) was conducted to check honey adulteration using the species-specific primers. There were 100 ng of DNA used for the first PCR and 5 μL of PCR product used as a template DNA in the second PCR following the procedure described before. PCR products were analyzed in 2.5% agarose gel and sequenced by Macrogen (Daejan, Korea) using an ABI 3130xl capillary automated. All sequences were generated in both directions and the forward and reverse sequences were assembled in BIOEDIT v7.0.5.2 (Hall, 1999) to produce a consensus sequence for each sample and the assembled sequences generated in this study were used to confirm the identification through DNA barcoding and have been deposited in GenBank under accession numbers MW660861-MW660880.

-Data analysis

From melting curve analysis, melting temperatures for 3 species of honey bees were compared by one-way analysis of variance (ANOVA) followed by Tukey’s post-hoc test. *p* values less than 0.05 were considered to be statistically significant. The statistical analysis was conducted using The R project software version 4.0.5 [32].

## 3. Results

### 3.1. Specificity Test of Species-Specific Primers

The DNA extracts from different honeybees with different geographical origins were used to examine the specificity of the designed species-specific primers. Each primer set successfully amplified *ND2* region from the DNA samples extracted from *A. cerana*, *A. dorsata,* and *A. mellifera* with an amplicon size of 224, 302, and 377 bp, respectively (Figure 2). None of the non-specific DNA amplification was observed with tested primer sets, suggesting these three species-specific primers can be successfully used to distinguish the origin of the honeybee at the DNA level.

### 3.2. Sensitivity Test of Primers Using Bee and Honey DNA

To evaluate the sensitivity of PCR-based assay, DNA samples from 3 different bees (*A. cerana*, *A. dorsata* and *A. mellifera*) were serially diluted (100 to 0.01 ng/μL) and used for PCR. The PCR condition using primer sets showed that all primer sets are able to amplify the specific bands (Figure 3). From the *A. dorsata* DNA, the AD-F/AD-R primer set successfully amplified the band with the expected size (302 bp) (Figure 3A). The intensity of characteristic bands was gradually raised as the concentration of DNA template increased, and the band could be visible when the DNA template was as low as 0.1 ng. From the *A*. *mellifera* (Figure 3B) and *A. cerana* DNA, similarly, AM-F/AM-R and AC-F/AC-R primer sets were also able to amplify specific bands with a detection limit of total 0.1 ng template DNA in the PCR reaction. This suggests that our species-specific primers can be used to detect the origin of honeybee samples with a low amount of DNA.

In other to test the ability to detect the target DNA among pure and adulterated honey samples, AMH was mixed with either ADH or ACH in different proportions. DNA was extracted from pure and mixed honey and used for the subsequent PCR analysis. Importantly, although the same amount of DNA (100 ng) from honey and bee samples was used for PCR, we were not able to detect the specific band from the first round of PCR with DNA from honey, unlike with DNA from bees (Figure 2 and Figure 3). This is likely due to actual amount of bee DNA being lower in DNA extracted from honey, as the honey sample contains biological tissues of other organisms (e.g., plant, microorganism, and other insect tissues). Thus, we performed another round of PCR by using 5 μL of PCR product as a template for analyzing the honey samples. In the second round of PCR, AC-F/AC-R and AD-F/AD-R primer sets produced a single band at the expected size with DNA from 100% ACD (Figure 4A, lane 1) and 100% ADH (Figure 4B, lane 1), respectively. On the contrary, the AM-F/AM-R primer set failed to amplify the band from the DNA extracted from 100% ACH (Figure 4A, lane 2) or 100% ADH (Figure 4B, lane 2). Neither AC-F/AC-R nor AD-F/AD-R primer sets amplified the specific bands from DNA extracted from 100% AMH (Figure 4A,B, lane 11).

DNA form ACH and ADH with different concentrations of AMH were also tested with species-specific primers (Figure 4A,B, lanes 3–12). In both conditions, the species-specific band of for *A. mellifera* was visible when the concentration of AMH was as low as 1%. The intensity of *A. mellifera* species-specific band gradually increased when the DNA from mixed honey with a proportion of 1 to 50% AMH were used and remained constant up to 100% AMH. To examine the possibility of using species-specific primer sets in the practical adulteration assay, 20 honey samples labeled as ADH, ACH, and AMH from different localities were tested. Analysis of the sequences of PCR products indicated that the primer sets are specific enough to detect the entomological origin of honey from different geographical localities.

### 3.3. Melting Curve Analysis by Real-Time PCR

To evaluate the possibility of use of melting curve analysis for detecting ACH or ADH adulteration, a real-time PCR experiment was conducted using the same PCR condition and primer sets and DNA extracted from honeybees. The result was confirmed using agarose gel electrophoresis and sequencing. Melting curve analysis of real-time PCR products demonstrated two distinct curves allowing the discrimination of *A. dorsata* from *A. mellifera* (Figure 5A) and of ADH from AMH. The melting temperature (Tm) of amplicons generated from *A. dorsata* (69.2 ± 0.1 °C) was distinct from the *A. mellifera* (72.4 ± 0.1 °C); hence, the detection of Tm could be an alternative method to detect the origin of ADH in addition to standard PCR method. Melting curve analyses of PCR products between *A. cerana* and *A. mellifera* were also performed (Figure 5B). Tm of amplicons of *A. cerana* (71.9 ± 0.2 °C) was distinct from *A. mellifera* (72.4 ± 0.1 °C) but very similar; hence, the use of Tm for distinguishing *A. cerana* and *A. mellifera* need more caution. The results of one-way ANOVA indicated that there was a significant difference between the Tm values of all three species (F value = 325.2, *p*-value < 0.001).

## 4. Discussion

Although previous attempts based on DNA barcoding of 16S rRNA and *COI* genes were helpful to inspect mislabeling [33], it was not functional to detect honey adulteration. In spite of the availability of species-specific primers to differentiate ACH from AMH [30], the primers developed by Kim et al. [30] were only applicable to honey originated from Korea but failed to differentiate ACH and AMH originated from China [26]. Since Soares et al. [31] only developed species-specific primers (AC1-F/AC1-R) to amplify 111 bp of tRNAlux-cox2 intergenic region of *A. cerana* mtDNA, the new *A. mellifera* species-specific primers were needed for the adulteration test. In addition, although AC1-F/AC1-R primers were useful in the discrimination of ACH from AMH, unlike ACF2/ACR2, they also amplified the non-specific band from ADH DNA extract, suggesting AC1-F/AC1-R was not enough to distinguish ACH from ADH (Appendix A). In this study, we provided not only the first species-specific primer set to identify ADH, but also two new species-specific primer sets to identify ACH and AMH. Notably, our newly designed primers successfully amplified specific bands only from the targeted DNA sample and were able to discriminate both ACH and ADH from AMH, and vice versa, thus providing new DNA-based assay for testing entomological origin of honey (refer to Figure 4A,B).

Zhang et al. [26] developed a gDNA-based method for the identification of two different major honeys from domesticated honeybees in the market. Two sets of primers (C-F/C-R for *A. cerana* and M-F/M-R for *A. mellifera*) were designed to amplify the Major royal jelly protein 2 (MRJP2) gene, resulting in the different size of PCR product in the gel electrophoresis, making it useful to discriminate ACH from AMH. In addition, it is possible to identify the honey samples through real-time PCR-based Tm analysis. Although the predicted size of the PCR product was 212 bp for *A. cerana* and 393 bp for *A. mellifera*, but the length of amplicons for *A. mellifera* was 560 bp from the PCR, as the primers were designed based on complementary DNA (cDNA) without an intron. ACH and AMH samples were distinguishable using C-F/C-R and M-F/M-R primer sets; however, 560 bp tends to be long for accurate honey identification and adulteration test with relatively old honey samples, which can possibly have DNA degradation problems. Honey is a complex matrix, and its phenolic/H_2_O_2_ induced oxidative stress would lead the DNA that remained inside honey to be easily degraded as storage time increased [31,34,35,36]. Notably, Schnell et al. [37] reported the diminished rate of successful amplification of amplicon size larger than 380 bp and in the fragmented DNA, thus the amplification of short amplicons is preferable [38]. Thus our new species-specific primer sets with amplicon size ranging 224~377 bp (refer to Figure 2 and Figure 3) would provide a better chance to successfully examine the old honey samples. Although the speed of degradation of DNA inside honey is not well understood and very difficult to predict accurately as different honey have different biochemical compositions, DNA degradation problem needs to be considered while examining the entomological origin of honey via DNA-based assay.

Real-time PCR-based identification of the entomological origin of honey was successfully developed previously to discriminate ACH and AMH using species-specific primers [26,31]. Refer to the Figure 5, the primer sets developed in the present study can be used to differentiate the entomological origin of three different types of honey. Although ADH and AMH can be simply differentiated using melting curve analysis, this method should be applied to differentiate ACH and AMH with caution due to the close Tm of the amplicons. Tm-based identification method is quick and accurate without the requirement of the gel electrophoresis step. Thus, it will provide a possible high-throughput analyses method for the identification of the origin of honey.

The cost of conducting analysis for one honey sample using the combination of two-round PCR and subsequent gel electrophoresis using the methodology described in current research is 6.4$ per honey sample (DNA extraction kit, PCR master mix, agarose powder, ladder, TBE buffer, staining dye and primer cost), however, these expenses for authentication analysis using Real-Time PCR technic is about 5.2$ (DNA extraction, master mix, and primer cost). The electricity and labor cost required to run the equipment have not been considered in our calculation. Although the cost of the authentication analysis per sample is slightly lower using Real-Time PCR, it is more expensive to establish such facilities in comparison to the conventional PCR method. Furthermore, according to our calculations, the duration of analysis using Real-Time PCR is slightly longer (~35 min) than conventional PCR.

## 5. Conclusions

Three species-specific primer sets targeting the *NADH dehydrogenase 2* (*ND2*) region of mtDNA were designed and successfully applied to trace the entomological origin of honey produced by different honeybees, *A. cerana*, *A. dorsata* and *A. mellifera*. In addition, the *A. mellifera* specific primer set is applicable in honey fraud detection. The possibility of using melting curve analysis in discrimination of the origin of honey using the same primer sets is also confirmed. Our preliminary studies indicated the impossibility of providing species-specific primers with a smaller size of PCR product in the mitochondrial DNA (except the one provided by Soares et al. [7] for ACH). However, further studies targeting nuclear DNA are required. PCR-based method using species-specific primers provides a rapid and cost-effective method to screen the entomological origin of honey. Therefore, the development of new primer sets to identify honey produced by other species of honeybees will be valuable. On the other hand, more studies are needed to understand the pace of DNA degradation in honey and the applicability and limitations of using molecular methods in the authentication of older honey samples.

## Figures and Tables

**Figure 1 foods-11-00928-f001:**
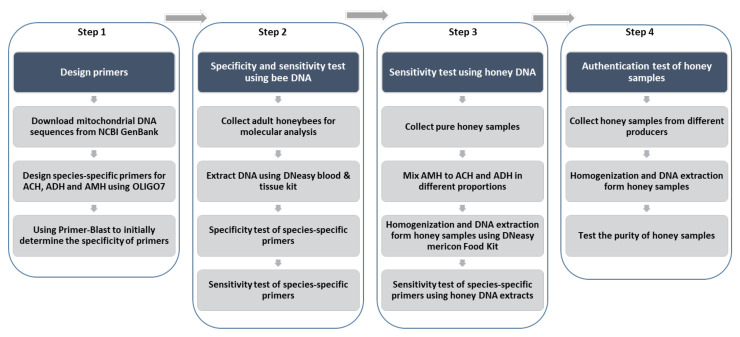
Schematic overview of the experimental design.

**Figure 2 foods-11-00928-f002:**
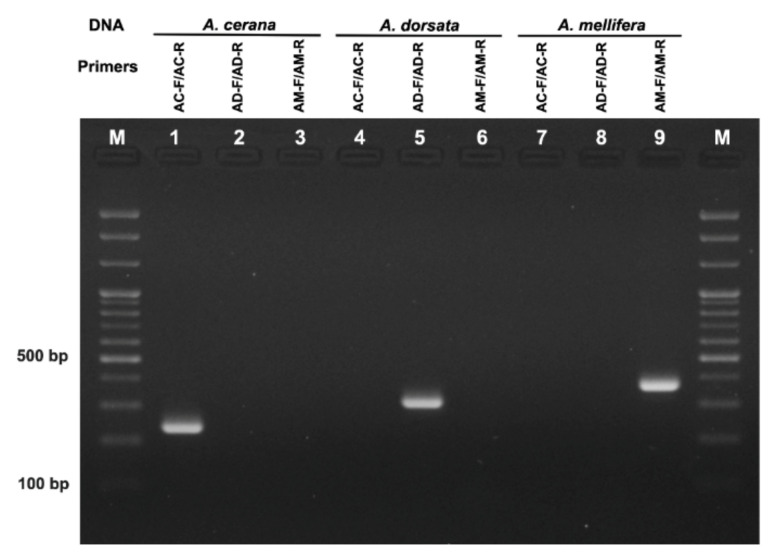
Agarose gel electrophoresis of PCR products amplified from DNA extracted of honeybees with species-specific primers. Bee DNA (Lanes 1–3: *A. cerana* DNA, Lanes 4–6: *A. dorsata* DNA, Lanes 7–9: *A. mellifera* DNA) Primers (Lanes 1, 4 and 7: *A. cerana* specific primers AC-F/AC-R, Lanes 2, 5 and 8: *A. dorsata* specific primers AD-F/AD-R, Lanes 3, 6 and 9: *A. mellifera* specific primers AM-F/AM-R), M: 100 bp ladder.

**Figure 3 foods-11-00928-f003:**
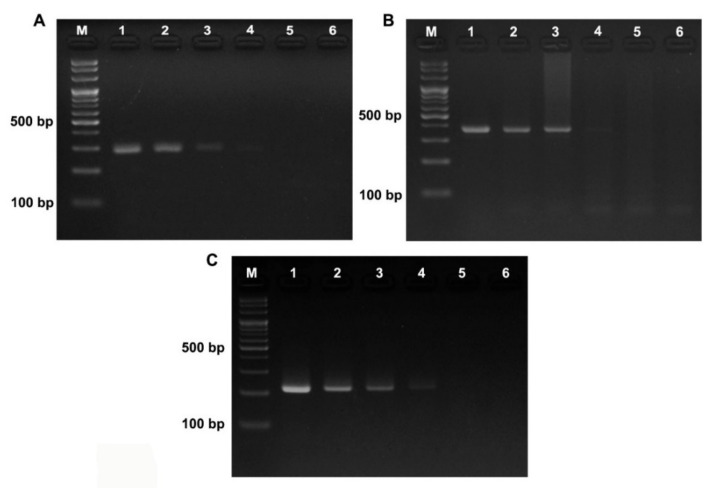
Sensitivity test of the designed species-specific primers using serially diluted DNA extract of *A. dorsata* (**A**), *A. mellifera* (**B**), and *A. cerana* (**C**). Lane M, DNA marker; Lane 1, 100 ng; lane 2, 10 ng; lane 3, 1 ng; lane 4, 0.1 ng; lane 5, 0.01 ng; lane 6, negative control.

**Figure 4 foods-11-00928-f004:**
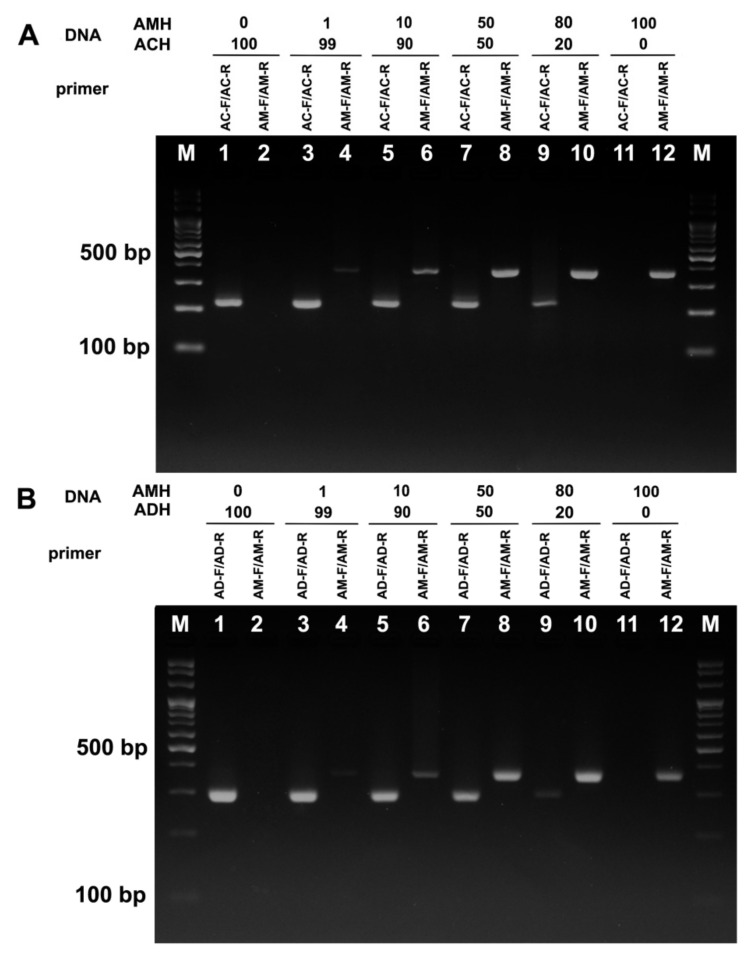
Adulteration test with artificially mixed honey samples. PCR products amplified from DNA extracted either from mixtures of ACH and AMH (**A**) or ADH and AMH (**B**) were analyzed by DNA gel electrophoresis. The proportions of AMH inside either ACH or ADH and usage of species-specific primer sets are shown. AC-F/AC-R, *A. cerana* specific primers; AD-F/AD-R, *A. dorsata* specific primers; AM-F/AM-R, *A. mellifera* specific primers; M, 1 kb ladder.

**Figure 5 foods-11-00928-f005:**
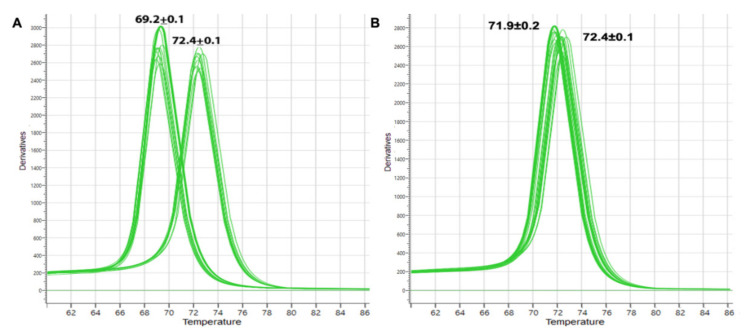
Conventional melting curves obtained by real-time PCR amplification targeting *ND2* region of mtDNA using DNA extracts from honeybees. (**A**) *A. dorsata* (Tm = 69.2 ± 0.1) and *A. mellifera* (Tm = 72.4 ± 0.1). (**B**) *A. cerana* (Tm = 71.9 ± 0.2) and *A. mellifera* (Tm = 72.4 ± 0.1).

**Table 1 foods-11-00928-t001:** Specific-primers for honey identification, nucleotide sequence, primer length, and the expected length of PCR product.

Species	Primer	5′-3′	Length	Target Fragment
*A. cerana*	AC-F	TCATTAGATTTTACAAAATCAGATCA	26	224 bp
	AC-R	CTTATAACTAAATATGTTAATGATCATA	28	
*A. dorsata*	AD-F	TATATTAATTGTTATAACTTACATAAATAA	31	302 bp
	AD-R	GGATTAAGAATATATAATATTCATATTTT	29	
*A. mellifera*	AM-F	CTATTAGATTTACTAAAACAGATACT	26	377 bp
	AM-R	ATAATTAAATGAATATAAAATAATTATAGCA	31	

## Data Availability

All the sequences generated in this research, have been deposited in GenBank under accession numbers MW660861-MW660880.

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
