# Peer review of "DNA-Based Method for Traceability and Authentication of Apis cerana and A. dorsata Honey (Hymenoptera: Apidae), Using the NADH dehydrogenase 2 Gene"

_foods, 2022, doi:10.3390/foods11070928_

Round 1

Reviewer 1 Report

The study designed three species-specific primers, targeting NADH dehydrogenase 2 16 (ND2) region of honeybee mitochondrial DNA, which were tested to distinguish the ento-17 mological origin of ACH, ADH and AMH. The study is very promising and can be improved further based on the following:
a) the introduction is ok, but also needs more information , there should be a paragraph on the importance of traceability and authentication in honey production, and why molecular detection methods are appropriate. This should be paragraph three, before line 61/62 begins. 
Please check line 79 [reference appears repeated] [27-2929]....line 86, reference Soares et al. (2018)...[x] for consistency
Please, try to strengthen the justification further...why is this work necessary, what would be the potential implication of conducting such a study?

Materials and methods is ok. It can be improved further by starting it with a subsection called 'Schematic overview of the experimental program' which should comprise 2-3 sentences, and a flow diagram describing the pathway, from the identification of honey sources (show them in boxes, then link them to the next stage with arrow)>collection of honey samples> harvest procedures> process of DNA extraction> etc etc ...Please, this is compulsory to help guide readers to follow your methods

Next subsection should then be:  "identification of honey sources", the next should then be "collection of honey samples", so please authors, kindly rearrange the beginning of materials and methods to adhere to these steps, provide more information, the reviewer would like to see a more elaborate sub-sections.

Please provide the GRPS of the sampling locations...it is very important to help authenticate and validate the source of the samples

Statistical analysis section is missing? You had some data, and you performed some analysis, kindly provide the statistical details you used ok

results and discussion...is very ok...the only part that needs more work is the discussion...firstly, please, insert (Refer to Figure x) in the discussion where specific results shown in one figure or another is being discussed/referred to. That is to say that the discussion must capture all the figures that have been mentioned in the results section.

Please, authors should extract more key aspects of the results and ensure to link it better with the relevant literature being used to make the argument/case.

Please, kindly provide a conclusion section, please ensure it concludes the work, provide what is the take-home message, it should stand clearly different from the abstract. Please, provide some direction for future studies. Thank you very much.

Author Response

Reviewer 1:

Comments and Suggestions for Authors

The study designed three species-specific primers, targeting NADH dehydrogenase 2 16 (ND2) region of honeybee mitochondrial DNA, which were tested to distinguish the entomological origin of ACH, ADH and AMH. The study is very promising and can be improved further based on the following:

Comment: The introduction is ok, but also needs more information, there should be a paragraph on the importance of traceability and authentication in honey production, and why molecular detection methods are appropriate. This should be paragraph three, before line 61/62 begins. 

Answer: Thank you for your comment. Although this information was available in the manuscript, we misplaced it at the beginning of the introduction part. We transferred those sentences to the place that the reviewer had been proposed. In addition the appropriateness of using molecular detection is discussed in the next paragraph : ” Molecular detection of the entomological origin of honey by using the set of specific-primers is regarded as a rapid, accurate and suitable tool for identification of origin of animal products and processed foods [20-23]. Considering the method of processing honey by honeybees, the bee cells can be remained inside of honey. Given the opportunity to extract bee DNA from honey, it is possible to use it for identification of entomological origin of honey. Compared to the other identification methods for entomological origin of honey, such as SDS-PAGE or chemical-based methods [18, 24-25], DNA-based method is more precise, quick and suitable for analysis of large sample size [26].”

Comment: Please check line 79 [reference appears repeated] [27-2929]....line 86, reference Soares et al. (2018)...[x] for consistency

Answer: The format of reference was corrected accordingly, however we can not find the repetitious numbers in the “word” format of the paper and it seems it happened during converting it into “PDF” format. We will discuss with the journal about this problem.

Comment: Please, try to strengthen the justification further...why is this work necessary, what would be the potential implication of conducting such a study?

Answer: Thank you for your comment. It has been done accordingly

Comment: Materials and methods is ok. It can be improved further by starting it with a subsection called 'Schematic overview of the experimental program' which should comprise 2-3 sentences, and a flow diagram describing the pathway, from the identification of honey sources (show them in boxes, then link them to the next stage with arrow)>collection of honey samples> harvest procedures> process of DNA extraction> etc etc ...Please, this is compulsory to help guide readers to follow your methods

Answer: We provided a schematic diagram describing the pathway of material and method part.

Comment: Next subsection should then be:  "identification of honey sources", the next should then be "collection of honey samples", so please authors, kindly rearrange the beginning of materials and methods to adhere to these steps, provide more information, the reviewer would like to see a more elaborate sub-sections.

Answer: The material and methods section has been completely re-arranged with more elaborated sub-sections as recommended

Comment: Please provide the GRPS of the sampling locations...it is very important to help authenticate and validate the source of the samples

Answer: The co-ordinations have been added to the supplementary table 1.

Comment: Statistical analysis section is missing? You had some data, and you performed some analysis, kindly provide the statistical details you used

Answer: Answer: The statistical analysis of the Tm values of all three species of honeybees was conducted and added as a statistical analysis section.

Comment: Results and discussion...is very ok...the only part that needs more work is the discussion...firstly, please, insert (Refer to Figure x) in the discussion where specific results shown in one figure or another is being discussed/referred to. That is to say that the discussion must capture all the figures that have been mentioned in the results section.

Answer: We referred all figures in the discussion part as requested.

Comment: Please, authors should extract more key aspects of the results and ensure to link it better with the relevant literature being used to make the argument/case. Please, kindly provide a conclusion section, please ensure it concludes the work, provide what is the take-home message, it should stand clearly different from the abstract. Please, provide some direction for future studies. Thank you very much.

Answer: We have provided the conclusion section including the knowledge gap which worth to fill in future studies.

Reviewer 2 Report

The article entitled DNA-based method for traceability and authentication of Apis cerana and A. dorsata honey (Hymenoptera: Apidae), using the NADH dehydrogenase 2 gene  is an interesting one and need a minor correction.

Improve the quality of fig 1 and fig 2

I would recommend a PCA for the samples analysed.

Have you tried too

Author Response

Reviewer 2:

Comment: The article entitled DNA-based method for traceability and authentication of Apis cerana and A. dorsata honey (Hymenoptera: Apidae), using the NADH dehydrogenase 2 gene  is an interesting one and need a minor correction.

Answer: Thank you for your comments

Comment: Improve the quality of fig 1 and fig 2

Answer: Thanks for your comments, We provided better quality figure 1 however we did not have any better picture for Figure 2 to replace.

Comment: I would recommend a PCA for the samples analysed.

Have you tried too

Answer: We believe that the Principle Component Analysis is not appropriate in our study where we do not have many variables. We designed three sets of species-specific primers and tried to find out whether they are specific and sensitive enough to be applied in the traceability of the entomological origin of honey samples.

Reviewer 3 Report

Eventhough the manuscript submitted to Foods provides some novel insights regarding the entomological and botanical origin identification of honey (or geographical origin), the construction and interpretation of data is poor. The authors firstly must improve thoroughly the English language and the technical criteria of the article. There are important missing sections within the article such as Statistical analysis section, Conclusion section, etc. I enclose within the attached pdf comments for authors to improve their manuscript. Discussion on other authentication techniques of honey must be provided for comparison.

Based on these comments, I suggest a major revision prior to further consideration for publication.

Author Response

Reviewer 3:

Comment: Eventhough the manuscript submitted to Foods provides some novel insights regarding the entomological and botanical origin identification of honey (or geographical origin), the construction and interpretation of data is poor. The authors firstly must improve thoroughly the English language and the technical criteria of the article. There are important missing sections within the article such as Statistical analysis section, Conclusion section, etc. I enclose within the attached pdf comments for authors to improve their manuscript. Discussion on other authentication techniques of honey must be provided for comparison.

Based on these comments, I suggest a major revision prior to further consideration for publication.

Answer: Thank you for your comments. The grammar errors were removed from the manuscript. In this manuscript, we have only provided a methodology to identify the entomological origin of honey for three different honeybee species and we did not study the BOTANICAL or GEOGRAPHICAL origin of the honey samples.The format of reference was corrected accordingly, however we can not find the repetitious numbers in the “word” format of the paper and it seems it happened during converting it into “PDF” format. We will discuss with the journal about this problem. The material and method section is re-arranged for better understanding of readers. We have also provided a schematic diagram describing the pathway of material and method part.

Comment: The authors must provide the cost of these DNA methodologies and comparision with other authentication technoques of honey botanical origin.

Answer: Thank you for your valuable and important comment. Although the prices can be fluctuated over a time, we provided the expenses which is required to analyze each sample using PCR and real-time PCR. The cost comparison with other authentication technics is not given due to lack of our knowledge about the details of the expenses required using other technologies.

Comment: Statistical analysis section is missing. Statistical analysis of the obtained data and classification models must be given

Answer: The statistical analysis of the Tm values of all three species of honeybees was conducted and added as a statistical analysis section.

Comment: Conclusion is missing

Answer: We have provided the conclusion section highlighting the important results of our scientific manuscript. We also included the knowledge gap which worth to fill in future studies.

Round 2

Reviewer 1 Report

Thanks to the authors for the very strong revising that has been performed.
Please, a few areas were noticed.
Line 345: (Refer to Fig. 4A, B)
Line 362: (Refer to Fig. 2-3)
Please check others in the discussion to ensure it has 'Refer to Fig.??
Also, kindly go through the discussion to ensure that all figures mentioned in the results section are captured/covered
Line 378-379 (Please, provide a reference point to the cost, if it is current cost, indicate where this cost is got from, and on what is based on)
Line 396: more studies (not 'studied') (I guess it is typo error)

Please, further brainstorm on the conclusions ok
After all these, I believe it is acceptable for publication

Author Response

Thanks to the authors for the very strong revising that has been performed.
Please, a few areas were noticed.

Comment: Line 345: (Refer to Fig. 4A, B)

Answer: It is done accordingly

Comment: Line 362: (Refer to Fig. 2-3)

Answer: It is done accordingly

Comment: Please check others in the discussion to ensure it has 'Refer to Fig.??

Answer: It is done accordingly

Comment: Also, kindly go through the discussion to ensure that all figures mentioned in the results section are captured/covered

Answer: It is done accordingly

Comment: Line 378-379 (Please, provide a reference point to the cost, if it is current cost, indicate where this cost is got from, and on what is based on)

Answer: Thank you for your comment. We have included the details of the items which we have calculated the cost based on.

Comment: Line 396: more studies (not 'studied') (I guess it is typo error)

Answer: It is done accordingly

Comment: Please, further brainstorm on the conclusions ok

Answer: It is done accordingly

Reviewer 3 Report

The revised version is acceptable for publication. The authors must give all the honeybees species in italic. Check and correct through the proof-reading.

Author Response

The authors must give all the honeybees species in italic. Check and correct through the proof-reading.

Answer: It is done accordingly